# Association of Polymorphisms in *FSHR*, *INHA*, *ESR1*, and *BMP15* with Recurrent Implantation Failure

**DOI:** 10.3390/biomedicines11051374

**Published:** 2023-05-05

**Authors:** Eun-Ju Ko, Ji-Eun Shin, Jung-Yong Lee, Chang-Soo Ryu, Ji-Young Hwang, Young-Ran Kim, Eun-Hee Ahn, Ji-Hyang Kim, Nam-Keun Kim

**Affiliations:** 1Department of Biomedical Science, College of Life Science, CHA University, Seongnam 13488, Republic of Korea; ejko05@naver.com (E.-J.K.); smilee3625@naver.com (J.-Y.L.); regis2040@nate.com (C.-S.R.); 2Department of Obstetrics and Gynecology, Fertility Center of CHA Bundang Medical Center, CHA University, Seongnam 13520, Republic of Korea; 1219annie@cha.ac.kr (J.-E.S.); happyimam@naver.com (Y.-R.K.); bestob@chamc.co.kr (E.-H.A.); 3Department of Obstetrics and Gynecology, Fertility Center of CHA Gangnam Medical Center, CHA University, Seoul 06125, Republic of Korea; jyhwang@chamc.co.kr

**Keywords:** recurrent implantation failure, single-nucleotide polymorphism, genotype combination, female hormones

## Abstract

Recurrent implantation failure (RIF) refers to two or more unsuccessful in vitro fertilization embryo transfers in the same individual. Embryonic characteristics, immunological factors, and coagulation factors are known to be the causes of RIF. Genetic factors have also been reported to be involved in the occurrence of RIF, and some single nucleotide polymorphisms (SNPs) may contribute to RIF. We examined SNPs in *FSHR*, *INHA*, *ESR1*, and *BMP15*, which have been associated with primary ovarian failure. A cohort of 133 RIF patients and 317 healthy controls consisting of all Korean women was included. Genotyping was performed by Taq-Man genotyping assays to determine the frequency of the following polymorphisms: *FSHR* rs6165, *INHA* rs11893842 and rs35118453, *ESR1* rs9340799 and rs2234693, and *BMP15* rs17003221 and rs3810682. The differences in these SNPs were compared between the patient and control groups. Our results demonstrate a decreased prevalence of RIF in subjects with the *FSHR* rs6165 A>G polymorphism [AA vs. AG adjusted odds ratio (AOR) = 0.432; confidence interval (CI) = 0.206–0.908; *p* = 0.027, AA+AG vs. GG AOR = 0.434; CI = 0.213–0.885; *p* = 0.022]. Based on a genotype combination analysis, the GG/AA (*FSHR* rs6165/ESR1 rs9340799: OR = 0.250; CI = 0.072–0.874; *p* = 0.030) and GG-CC (*FSHR* rs6165/*BMP15* rs3810682: OR = 0.466; CI = 0.220–0.987; *p* = 0.046) alleles were also associated with a decreased RIF risk. Additionally, the *FSHR* rs6165GG and *BMP15* rs17003221TT+TC genotype combination was associated with a decreased RIF risk (OR = 0.430; CI = 0.210–0.877; *p* = 0.020) and increased FSH levels, as assessed by an analysis of variance. The *FSHR* rs6165 polymorphism and genotype combinations are significantly associated with RIF development in Korean women.

## 1. Introduction

Implantation is a process by which an embryo attaches to the lumen surface of the endometrium and then migrates into the deep layer of the endometrium [1]. Successful embryo implantation requires success at each event, such as sperm and oocyte quality, development of the early embryo and endometrium, and the interaction between the blastocyst and endometrium [2,3]. Recurrent implantation failure (RIF) is defined as repeated embryo implantation failure after four high-quality embryo transfers in women. Despite advances in in vitro fertilization (IVF) technology, success rates remain stable, and approximately 10% of women undergoing IVF treatment suffer from RIF [4]. Implantation failure can be a consequence of immunological, uterine, male, embryo, or coagulation factors and genetics [5,6]. Among them, embryogenesis and development are some of the most important processes in pregnancy [7]. In the IVF procedure, only embryos classified as healthy according to specific criteria are selected and transplanted into the uterus. However, a good or “transplantable” class of embryos may increase the success rate of transplantation and pregnancy [8,9]. The morphological criteria do not necessarily correlate with implantation success rates, and approximately 50% of healthy embryos fail to implant [10]. Additionally, several factors are involved in embryogenesis and development [11,12,13,14,15,16].

Maternal hormones comprise an important factor for successful implantation during pregnancy. In particular, ovarian hormones interact with signaling molecules, including cytokines and growth factors, to assist embryo implantation [17]. Additionally, hormone receptors are associated with the implantation process and pregnancy result, and changes in the hormone receptor expression and function may affect the pregnancy outcome [18,19,20].

One of the most important hormones is follicle-stimulating hormone (FSH). FSH binds to the FSH receptor (FSHR) expressed in follicular granulosa cells to induce estrogen production and secretion, which stimulates the growth and maturation of ovarian follicles and subsequently improves the quality and recovery rate of oocytes [21]. Previously, the FSHR was thought to be expressed only in the ovaries, but a recent study showed that it was expressed in extra-gonadal tissues including the cervix, endometrium, myometrium, vascular smooth muscle, and vascular endothelium [22]. One study confirmed that FSHR expression was upregulated during decidualization and in the myometrium during pregnancy. In addition, a study in mice showed that FSHR was expressed in the placental blood vessels and was related to fetal vascular formation, and deletion of the *FSHR* gene affected pregnancy [23,24]. Extraovarian FSHRs play an important role in establishing and maintaining successful pregnancies in humans [22]. Previously reported studies showed that FSHR was expressed in fetal vascular endothelium [23], and it was identified that the FSH–FSHR signaling system promotes the angiogenesis of vascular endothelial cells [25]. FSHR is also expressed in the uterine myometrium and plays an important role in regulating uterine muscle contraction [26]. The *INHA* gene encodes inhibin α, which suppresses FSHR expression, thereby inhibiting the action of FSH [27]. Inhibin α is expressed in the luminal epithelium, glandular epithelium, stromal tissue, and vascular endothelium throughout the menstrual cycle [28]. Inhibin α is secreted by mature follicles and reflects follicular maturity. One study showed that serum inhibin α is a predictor for determining oocyte maturation [29]. Furthermore, concentrations of inhibin α in follicular fluid were higher in the pregnancy group than in the non-pregnancy group, and a positive correlation was found between the number of oocytes retrieved and the fertility rate [30]. Estrogen receptors, ERα and ERβ, mediate well-characterized effects on follicle growth, maturation, oocyte release, and endometrial preparation for implantation by binding to estrogen [31]. Among them, ERα is encoded by estrogen receptor 1 (*ESR1*) and plays an essential role in regulating decidualization [32]. Bone morphogenetic protein (*BMP*) 15 is a TGFβ family member secreted by oocytes during follicle formation and is expressed in granulocytes and follicles as well as in oocytes. BMP15 plays an important role in follicular development during primordial follicle recruitment, ovulation, and corpus luteum formation and is closely related to fertilization, embryonic quality, and pregnancy outcomes. Therefore, it can be considered a new molecular marker for predicting follicular development potential [33]. In addition, one study showed that the concentration of BMP15 in women with an intracytoplasmic sperm injection was related to the fertility level, and an increased follicular concentration of BMP15 indicated an increased fertility level [34].

Single-nucleotide polymorphisms (SNPs) have been associated with reproductive diseases [35,36,37]. Additionally, RIF has previously been associated with SNPs, and many studies have been published exploring the relevance of these associations [38,39]. Polymorphisms in regulatory regions (promoter, 5′, 3′ UTR) or gene-coding regions may alter gene expression [40,41]. Maternal hormones play an important role in maintaining pregnancy, and hormone levels may be altered by certain polymorphisms [42,43]. Therefore, four genes (*FSHR*, *INHA*, *ESR1*, and *BMP15*) related to hormones were selected, and polymorphisms located in gene regulatory or coding regions were selected. Finally, a total of seven mutations were selected: coding region (*FSHR* rs6165, *ESR1* rs2234693, and *BMP15* rs17003221), promoter region (*INHA* rs11893842, rs35118453, and *ESR1* rs9340799), and 5′UTR region (*BMP15* rs3810682). The occurrence of *FSHR*, *INHA*, *ESR1*, and *BMP15* gene polymorphisms has also been reported to be associated with various reproductive diseases, including primary ovarian insufficiency (POI), pregnancy loss, and preeclampsia [44,45,46,47]. However, few studies have examined the associations between RIF and *FSHR*, *INHA*, *ESR1*, and *BMP15* gene polymorphisms. In this study, we investigated whether *FSHR*, *INHA*, *ESR1*, and *BMP15* polymorphisms are associated with RIF and whether these polymorphisms affect the levels of clinical factors in Korean women. To reveal the relationship between RIF and *FSHR*, *INHA*, *ESR1*, and *BMP15* gene polymorphisms, we assessed the differences between RIF patients and healthy controls by examining the known *FSHR* (rs6165), *INHA* (rs11893842, rs35118453), *ESR1* (rs9340799, rs2234693), and *BMP15* (rs17003221, rs3810682) gene polymorphisms.

## 2. Materials and Methods

### 2.1. Study Population

Blood samples were obtained from women with RIF treated at the Department of Obstetrics and Gynecology and the Fertility Center of CHA Bundang Medical Center in Seongnam, South Korea, between March 2010 and December 2022. In total, we obtained blood samples from 133 patients with RIF and 317 control participants. All patients and controls were Korean. The institutional review board of CHA Bundang Medical Center approved the study, and all patients provided written informed consent (reference no. CHAMC2009-12-120). All embryos were examined by an embryologist prior to transfer, and embryos that showed good quality were transferred. Implantation failure refers to cases where the level of human chorionic gonadotropin measured on the 14th day of embryo transfer is <5 U/mL [48]. In the study group, subjects diagnosed with implantation failure due to anatomical, chromosomal, hormonal, infectious, autoimmune, or thrombotic causes were excluded. Uterine anatomical abnormalities in RIF patients were confirmed by hysterosalpingography, hysteroscopy, uterine sonography, computed tomography, or magnetic resonance imaging. A karyotype analysis was conducted to confirm chromosomal abnormalities, and the karyotype analysis followed the standard protocol. Hormonal causes of RIF such as hyperproactinemia, lutein insufficiency, and thyroid disease were identified by measuring prolactin, thyroid stimulating hormone (TSH), free T4, FSH, luteinizing hormone (LH), and progesterone levels in peripheral blood. Lupus anticoagulants and anti-cardiolipin antibodies were tested to confirm the autoimmune disease lupus and antiphospholipid syndrome, respectively. A deficiency of protein C and protein S and the presence of anti-β2 glycoprotein antibodies were diagnosed as thrombosis. As a control group, women with regular menstrual cycles, pregnancy history of at least one naturally conceived pregnancy, no history of pregnancy loss, and karyotype of 46, XX were recruited from CHA Bundang Medical Center.

### 2.2. Estimation of Homocysteine, Folate, Total Cholesterol, Uric Acid, Blood Urea Nitrogen, Creatinine, and Blood Coagulation Status

Blood samples were collected from RIF subjects after 12 h of fasting. We performed a fluorescence polarization immunoassay using the Abbott IMx analyzer (Abbott Laboratories, Abbott Park, IL, USA) to measure the homocysteine level. Folate was measured via a competitive immunoassay using the ACS 180Plus automated chemiluminescence system (Bayer Diagnostics, Tarrytown, NY, USA). Total cholesterol, uric acid, blood urea nitrogen, and creatinine were measured using commercially available enzymatic colorimetric assays (Roche Diagnostics, GmbH, Mannheim, Germany). The platelet, white blood cell, and hemoglobin levels were obtained using the Sysmex XE 2100 automated hematology system (Sysmex Corporation, Kobe, Japan). The prothrombin time (PT) and activated partial thromboplastin time (aPTT) were measured with an ACL TOP automated photo-optical coagulometer (LSI Medience, Tokyo, Japan).

### 2.3. Flow Cytometry Analysis of Immune Cell Proportion

Immune cell measurement was performed by flow cytometry using CellQuest software (BD FACS Calibur; BD Biosciences, Seoul, Republic of Korea). Fluorescently-labeled [fluorescein isothiocyanate, phycoerythrin (PE), peridinin chlorophyll protein, and allophycocyanin] monoclonal antibodies specific to CD3, CD4, CD8, CD19, CD16, and CD56 were purchased from BD Biosciences. Anti-NKG2A-PE antibodies were obtained from Beckman Coulter (Fullerton, CA, USA). To determine cell surface antigen expression, peripheral blood mononuclear cells (2.5 × 105) were stained for 30 min at 4 °C in the dark, washed twice with 2% phosphate-buffered saline containing 1% bovine serum albumin and 0.01% sodium azide (FACS wash buffer), and then fixed with 1% formaldehyde solution (Sigma-Aldrich, St. Louis, MO, USA) prior to sorting, as previously described [49].

### 2.4. Hormone Assays

On the second to third days of the women’s menstrual cycle, blood samples were collected and measured using serum samples. Following the manufacturer’s instructions, the estradiol (E2) and TSH levels were measured using radioimmunoassays (Beckman Coulter), and the FSH and LH levels were measured using enzyme immunoassays (Siemens, Munich, Germany).

### 2.5. SNP Selection and Genetic Analysis

We selected *FSHR*, *INHA*, *ESR1*, and *BMP15*, which are hormone-related genes associated with pregnancy. To select polymorphisms of the *FSHR*, *INHA*, *ESR1*, and *BMP15* genes, studies on the association between pregnancy-related diseases (recurrent pregnancy loss, recurrent implantation failure, preeclampsia, premature ovarian failure, and poor ovarian response) and polymorphisms were investigated [46,50,51,52,53,54,55]. Finally, a total of seven polymorphisms in *FSHR* (rs6165), *INHA* (rs11893842 and rs35118453), *ESR1* (rs9340799 and rs2234693), and *BMP15* (rs17003221 and rs3810682) were selected and studied. Genomic DNA was extracted from anticoagulated peripheral blood using a G-DEX blood extraction kit (Intron, Seongnam, Republic of Korea). All genetic polymorphisms were identified by a real-time polymerase chain reaction (PCR) using the TaqMan SNP Genotyping Assay Kit (Applied Biosystems, Foster City, CA, USA). To validate the real-time analysis, DNA sequencing was performed on approximately 10~15% of the samples by random selection using an ABI 3730XL DNA Analyzer (Applied Biosystems). The concordance of the quality control samples was 100%.

### 2.6. Statistical Analysis

Regarding the clinical characteristics of participants, the differences between the categorical variables were analyzed using a chi-square test and continuous variables using an independent sample *t*-test. The data were presented as mean and standard deviations for continuous variables, and frequency and percentage for categorical variables. The statistical normality of continuous variables was confirmed using the Kolmogorov–Smirnov test. For continuous variables showing a non-normal distribution (*p* < 0.05 in the Kolmogorov–Smirnov test), group difference analyses were performed using the Mann–Whitney test. Allele frequencies were determined to confirm deviations from the Hardy–Weinberg equilibrium. To select the best inheritance model for a specific polymorphism, Akaike’s information criterion was calculated. The associations between *FSHR*, *INHA*, *ESR1*, and *BMP15* polymorphisms and RIF incidence were calculated using adjusted odds ratios (AORs) and 95% confidence intervals (95% CIs) obtained from a multivariate logistic regression analysis adjusted for age. False discovery rate correction was used to adjust multiple comparison tests and provide a measure of the expected proportion of false positives among the data. The open-source MDR software package (v.2.0, www.epistasis.org accessed on 1 April 2022) was used to perform the genetic interaction analysis. Using this MDR analysis, all possible genotype combinations for gene–gene interactions were identified and analyzed. The associations between each *FSHR*, *INHA*, *ESR1*, and *BMP15* gene polymorphism and each clinical value (platelets, PT, aPTT, homocysteine, folate, natural killer cells, uric acid, and total cholesterol) for RIF patients were assessed using ANOVA and Kruskal–Wallis tests. For the overall statistical analysis, the level of statistical significance was set as *p* < 0.05.

## 3. Results

Table 1 presents the clinical variables of the 133 RIF patients and 317 control subjects. No significant differences in age distribution were observed between the RIF and control groups, indicating that our age frequency matching was satisfactory. The blood urea nitrogen, creatinine, PT, TSH, E2, LH, BMI, total cholesterol, and white blood cell levels were different between the patient and the control group. The blood urea nitrogen (*p* < 0.0001), creatinine (*p* < 0.0001), PT (*p* < 0.0001), TSH (*p* = 0.0001), E2 (*p* = 0.0002), and LH (*p* < 0.0001) levels increased significantly in the patient group, and on the contrary, BMI (*p* = 0.047), total cholesterol (*p* < 0.0001), and white blood cell (*p* = 0.005) values increased significantly in the control group.

We investigated the distribution of *FSHR* (rs6165), *INHA* (rs11893842, rs35118453), *ESR1* (rs9340799, rs2234693), and *BMP15* (rs17003221, rs3810682) polymorphisms in RIF patients and the control group (Table 2). The AOR with respect to age was calculated from the logistic regression analysis. The frequency of each genotype in the control group was consistent with the Hardy–Weinberg equilibrium.

We investigated the genotype frequencies according to the number of implantation failures among RIF patients (Table 2). For *FSHR* rs6165A>G, GG homozygous genotypes and the recessive model were found to exert a protective effect against RIF [AA vs. GG: AOR, 0.463; 95% CI, 0.218–0.980; *p* = 0.044 and AA+AG vs. GG (recessive model): AOR, 0.430; 95% CI, 0.210–0.877; *p* = 0.020]. The other gene polymorphisms (*INHA* rs11893842 and rs35118453, *ESR1* rs9340799 and rs2234693, and *BMP15* rs17003221 and rs3810682) did not show significant differences between the control and patient groups. We also identified the genotype frequencies among patients according to the number of RIFs (Table 3). *FSHR* rs6165A>G GG genotypes and the recessive model showed protective effects in patients with more than three implantation failures (AA vs. GG: AOR, 0.412; 95% CI, 0.183–0.931; *p* = 0.033 and recessive model: AOR, 0.392; 95% CI, 0.156–0.982; *p* = 0.046), and *INHA* rs3511845C>T CT genotypes and CT+TT (dominant model) showed risk effects in patients with more than four implantation failures (CC vs. CT: AOR, 3.793; 95% CI,1.171–12.285; *p* = 0.026 and dominant model: AOR, 3.745; 95% CI, 1.176–11.921; *p* = 0.025).

We performed a genotype combination analysis of RIF patients and control subjects (Table 4). In the genotype combination type of *FSHR* rs6165 A>G and *ESR1* rs9340799 A>G, the GG/AA (OR, 0.250; 95% CI, 0.072–0.874; *p* = 0.030) and GG/AA+AG (OR, 0.373; 95% CI, 0.171–0.816; *p* = 0.014) combinations were significantly associated with RIF risk. Additionally, in the *FSHR* rs6165A>G/*INHA* rs35118453C>T combination genotype, GG/CC+CT (OR, 0.399; 95% CI, 0.190–0.841; *p* = 0.016) showed a significantly decreased OR. In contrast, in the genotype combination analysis of *INHA* rs35118453 and C>T/*ESR1* rs2234693 T>C, TT/TC+CC (OR, 7.001; 95% CI, 1.298–37.776; *p* = 0.024) was associated with an increased RIF risk. Additionally, in the *INHA* rs11893842 A>G and *ESR1* rs9340799 A>G genotype combination, AA+AG/GG (OR, 3.065; 95% CI, 1.081–8.690; *p* = 0.035) was associated with an increased RIF risk. In the genotype combination analysis of *ESR1* rs9340799 A>G/*ESR1* rs2234693 T>C, GG/TT+TC (OR, 12.930; 95% CI, 1.492–112.052; *p* = 0.020) was associated with an increased RIF risk.

We performed an analysis of the differences in clinical factors according to the genotype of the *FSHR* (rs6165), *INHA* (rs11893842, rs35118453), *ESR1* (rs9340799, rs2234693), and *BMP15* (rs17003221, rs3810682) polymorphisms using a one-way analysis of variance (ANOVA). We found that increased LH levels were associated with the *INHA* rs35118453 polymorphism among all subjects (Table 5, *p* < 0.05). In RIF patients, increased E2 levels were associated with *INHA* rs11893842, and decreased CD3 (pan T cell) levels were associated with *ESR1* rs2234693 (Appendix A, *p* < 0.05). TSH levels in the control group showed an increasing trend according to the FSHR rs6165 genotype (Appendix A, *p* < 0.05).

We then analyzed the synergistic effects of the *FSHR* (rs6165), *INHA* (rs11893842, rs35118453), *ESR1* (rs9340799, rs2234693), and *BMP15* (rs17003221, rs3810682) polymorphisms and clinical factors (i.e., folate, homocysteine, E2, FSH, and LH) on RIF risk (Appendix A). In Appendix A, we set the reference values for homocysteine, E2, FSH, and LH levels by determining the threshold for the highest 25% of levels (homocysteine 7.94 µmol/L, E2 37.6 pg/mL, FSH 9.6 U/L, and LH 5.12 U/L), and the folate level was determined by the lowest 25% of levels (8.12 ng/mL) in RIF patients and controls. *FSHR* rs6165 and high E2 and FSH levels had a synergistic effect on an increased susceptibility to RIF (AA+AG with high E2 levels, AOR = 11.411, *p* < 0.0001; AA+AG with high FSH levels, AOR = 3.009, *p* = 0.025, Figure 1). Additionally, *INHA* rs11893842, high homocysteine levels, and low folate levels had a synergistic effect on an increasing susceptibility to RIF (AG+GG with high homocysteine levels, AOR = 6.180, *p* = 0.004; AG+GG with low folate levels, AOR = 8.090, *p* = 0.003, Figure 1). Moreover, combining the *ESR1* rs9340799 AG+GG type with E2 (AOR = 6.538, *p* = 0.0001), FSH (AOR = 5.511, *p* = 0.003), and LH (AOR = 12.305, *p* = 0.002) resulted in an increased RIF risk (Figure 2). Furthermore, the AORs of the *ESR1* rs2234693 TC+CC group when combined with each female hormone factor including ≥37.6 pg/mL E2, ≥9.6 U/L FSH, and ≥5.12 U/L LH were 6.997 (*p* < 0.0001), 3.371 (*p* = 0.003), and 6.161 (*p* < 0.0001), respectively.

## 4. Discussion

The formation and quality of gametocytes and embryo development are important factors for successful pregnancy progression, and many infertile couples have experienced recurrent failure due to the poor quality or quantity of embryos before transplantation. Currently, morphological indicators are used for embryo selection, and there is no other predictive indicator that can evaluate the quality of embryo cells. Recently, many studies have elucidated molecular and genetic factors to assess embryo quality and select developmentally competent embryos. Previous studies have reported several genes associated with the pathogenesis of oocyte maturation arrest (*TUBB8*, *PATL2*) and fertilization failures (*TLE6*, *WEE2*). [56]. In this study, seven polymorphisms were selected from four hormone-related genes (*FSHR* rs6165, *INHA* rs11893842 and rs35118453, *ESR1* rs9340799 and rs2234693, *BMP15* rs1700321, and rs3810682), and the association between RIF occurrence and polymorphisms was analyzed.

We examined the association between RIF occurrence and SNPs in the *FSHR*, *INHA*, *ESR1*, and *BMP15* genes. Our results showed that the genotype frequency of the *FSHR* rs6165 SNP was significantly different in the control and RIF patient groups. A genotype combination analysis of the seven genetic markers revealed that the GG/AA (*p* = 0.030) and GG/AA+AG (*p* = 0.014) combinations significantly reduced the RIF risk in the *FSHR* rs6165 and *ESR1* rs9340799 combination type. Additionally, in the genotype combination analysis of *FSHR* rs6165A>G/*INHA* rs35118453C>T, GG/CC+CT (*p* = 0.016) showed a significantly decreased risk of RIF. In contrast, the *INHA* rs35118453 TT and *ESR1* rs2234693 TC+CC combination were associated with increased RIF risk (*p* = 0.024). Moreover, the *INHA* rs11893842 A>G/*ESR1* rs9340799 A>G, AA+AG/GG combination (*p* = 0.035) was associated with an increased RIF risk. In the genotype combination of *ESR1* rs9340799 A>G/*ESR1* rs2234693 T>C, GG/TT+TC (*p* = 0.020) was associated with an increased RIF risk. Furthermore, our gene–environment combinatorial analysis data revealed statistically significant relationships between the *FSHR* rs6165, *INHA* rs11893842, *ESR1* rs9340799, and rs2234693 genotypes and clinical factors (folate, homocysteine, E2, FSH, and LH) in Korean RIF patients. When combined with clinical parameters, the RIF risk increased by 2- to 11-fold.

Previously, *FSRH*, *INHA*, *ESR1*, and *BMP15* gene polymorphisms have been studied with regard to the conditions of the female reproductive system, including polycystic ovary syndrome, poor ovarian response, POI, breast cancer, and endometriosis, as well as male infertility [50,57,58,59,60,61,62,63,64,65,66,67].

The *FSHR* rs6165 polymorphism is located in exon 10 of the *FSHR* gene and is in the transmembrane region of the FSHR protein. The rs6165 polymorphism produces a change of A to G in position 919 and changes codon 307 from threonine (ACT) to alanine (GCT) [57]. According to Ganesh et al. [50], the *FSHR* rs6165 polymorphism was associated with unsuccessful IVF outcomes, and a higher frequency of the heterozygous AG genotype was observed in the infertile group than in the control group. For *FSHR* rs6165 AA carriers, the number of oocytes retrieved was significantly higher and ovarian stimulation was significantly shorter than those in GG and AG carriers [58]. Additionally, Rod et al. [59] found that *FSHR* rs6165 was associated with controlled ovarian stimulation and was 3-fold higher in poor responders than in good responders. Moreover, one study showed that the rs6165 AG genotype was associated with an increased risk of male infertility [60].

*INHA* rs11893842 and rs35119453 are located in the promoter region of the *INHA* gene and have been studied with regard to POI, male infertility (sperm parameters), and adrenocortical cancer [44,61,62]. Neither variant was associated with POI. Rafaqat et al. [48] found that the GG genotype frequency was increased in male infertility patients and showed a significant association with male infertility in the Pakistani population. Furthermore, rs11893842 minor alleles showed a low frequency in adrenocortical cancer patients and the rs11893842 AA genotype was associated with decreased INHA mRNA levels [62].

*ESR1* rs9340799 and rs2234693 are located in intron 1, 351 and 1397 bp upstream of exon 2, respectively. The two polymorphisms are associated with cancer in females, including breast cancer and endometrial cancer [63,64], and are associated with POI [65]. The *ESR1* rs9340799 GA and rs2234693 TC genotypes were associated with a decreased risk of POI, and the *ESR1* rs9340799 AA and rs2234693 TT genotypes were associated with an increased risk of POI [65]. Additionally, the *ESR1* rs9340799 GG genotype was associated with a 4-fold increased risk of endometriosis and a 3-fold increased risk of IVF failure in infertile patients [66]. Furthermore, the *FSHR* rs6165 GG, rs6165 AA, *ESR1* rs9340799 GA, and rs2234693 TC gene combination enhanced the protective effect of *FSHR* gene variants and was associated with a reduced risk of fibrocystic mastopathy in infertile women [67].

Several clinical factors are involved in embryo implantation and pregnancy maintenance, including folate, homocysteine, and female hormones (E2, FSH, LH). Gaskins et al. [68] found that, among women receiving assisted reproductive technology, women with high serum folate levels (>26.3 ng/mL) had a 1.62-fold higher live birth rate than women with low folate levels (<16.6 ng/mL). Additionally, Ocal et al. [69] found that high homocysteine levels in follicular fluid resulted in reduced cell division and increased fragmentation in embryo cultures, which was associated with decreased oocyte and embryo quality. Another study showed that exposure to high E2 concentrations is detrimental to blastocyst implantation and early post-implantation development. Moreover, during clinically-assisted reproductive technology, high serum E2 levels not only affected the endometrium but also directly affected the blastocyst during implantation [70]. Our study revealed that patients with several clinical factors (low folate levels and high homocysteine, E2, FSH, and LH levels) and *FSHR* rs6165, *INHA* rs11893842, *ESR1* rs9340799, and rs2234693 polymorphisms had an approximately 3- to 12-fold increased risk of RIF.

This study has several limitations. First, how *FSHR*, *INHA*, *ESR1*, and *BMP15* polymorphisms influence RIF development is still unclear. The effect of these SNPs should also be confirmed through in vitro and in vivo studies. Second, additional environmental risk factors to RIFs need to be evaluated. Finally, the size of the RIF patient and control group was small, and this study group included only Koreans. To determine whether the studied genetic polymorphism can be used as a predictor of RIF, our results should be validated using larger sample sizes and other ethnic groups.

## 5. Conclusions

We investigated the association between the *FSHR* rs6165, *INHA* rs11893842 and rs35118453, *ESR1* rs9340799 and rs2234693, and *BMP15* rs17003221 and rs3810682 polymorphisms and the risk of RIF in Korean women. We found the frequency of the GG genotype of *FSHR* rs6165 was lower among RIF patients than among controls, suggesting this genotype may be associated with a reduced risk of RIF. Additionally, the interaction of the *FSHR* rs6165, *INHA* rs11893842, and *ESR1* rs9340799 and rs2234693 polymorphisms with some clinical factors may increase the risk of RIF.

## Figures and Tables

**Figure 1 biomedicines-11-01374-f001:**
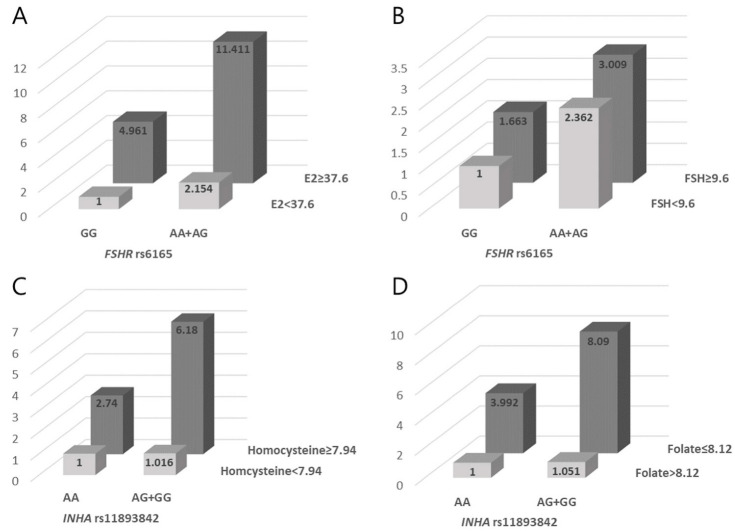
Synergistic effect of *FSHR* rs6165 and *INHA* rs11893842 polymorphisms with clinical parameters. (**A**–**D**) panels show AOR of *FSHR* rs6165 and *INHA* rs11893842 with clinical parameters including E2 (**A**), FSH (**B**), homocysteine (**C**), and folate (**D**).

**Figure 2 biomedicines-11-01374-f002:**
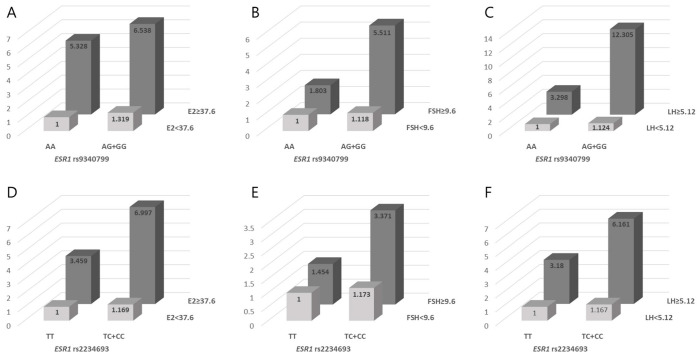
Synergistic effect of ESR1 rs9340799 and rs2234693 polymorphisms with clinical parameters. (**A**–**F**) panels show AOR of ESR1 rs9340799 and rs2234693 with clinical parameters including E2 (**A**,**D**), FSH (**B**,**E**), and LH (**C**,**F**).

**Table 1 biomedicines-11-01374-t001:** Clinical profiles between RIF patient and control subjects.

Characteristics	Controls (*n* = 317)	RIF (*n* = 133)	*p* ^a^
Age (years, mean ± SD)	33.5 ± 3.4	33.6 ± 2.9	0.579
BMI (kg/m^2^, mean ± SD)	22.0 ± 3.4	21.1 ± 3.2	0.047
Previous implantation failure (*n*, mean ± SD)	-	4.7 ± 2.0	N/A
Live births (*n*, mean ± SD)	1.6 ± 0.6	-	N/A
Mean gestational age (weeks, mean ± SD)	39.3 ± 1.6	-	N/A
Homocysteine (μmol/L mean ± SD)	6.4 ± 3.0	6.9 ± 1.8	0.402
Folate (ng/mL, mean ± SD)	14.0 ± 7.5	15.3 ± 10.7	0.896
BUN (mg/dL, mean ± SD)	8.8 ± 2.8	10.4 ± 2.9	<0.0001
Creatinine (mg/dL, mean ± SD)	0.6 ± 0.2	0.8 ± 0.1	<0.0001
Uric acid (mg/dL, mean ± SD	3.9 ± 1.0	4.0 ± 1.0	0.266
Total cholesterol (mg/dL)	215.3 ± 57.2	185.2 ± 42.6	<0.0001
WBC(10^3^/μL, mean ± SD	7.9 ± 2.2	7.4 ± 3.0	0.005
Hgb (g/dL, mean ± SD)	12.3 ± 1.2	12.4 ± 1.5	0.080
PLT (10^3^/μL)	230.0 ± 63.1	238.7 ± 67.0	0.465
PT (sec)	10.7 ± 1.6	11.3 ± 0.6	<0.0001
aPTT (sec)	29.1 ± 3.5	29.6 ± 3.4	0.164
CD3 (pan T) (%, mean ± SD)	-	66.9 ± 11.2	N/A
CD4 (helper T) (%, mean ± SD)	-	34.4 ± 8.8	N/A
CD8 (suppressor) (%, mean ± SD)	-	28.9 ± 7.7	N/A
CD19 (B cell) (%, mean ± SD)	-	11.7 ± 4.7	N/A
CD56 (NK cell) (%, mean ± SD)	-	18.1 ± 9.4	N/A
TSH (mU/L, mean ± SD)	1.6 ± 0.9	2.3 ± 1.5	0.0001
E2 (pg/mL, mean ± SD)	26.6 ± 14.4	33.3 ± 15.4	0.0002
FSH (U/L, mean ± SD)	8.1 ± 2.8	8.9 ± 4.5	0.599
LH (U/L, mean ± SD)	3.7 ± 2.6	4.8 ± 2.1	<0.0001

Note: RIF, recurrent implantation failure; BMI, body mass index; BUN, blood urea nitrogen; WBC, white blood cell; Hgb, hemoglobin; PLT, platelet count; PT, prothrombin time; aPTT, activated partial thromboplastin time; TSH, thyroid stimulating hormone; E2, estradiol; FSH, follicle stimulating hormone; LH, luteinizing hormone; N/A, not applicable. ^a^ Mann-Whitney test.

**Table 2 biomedicines-11-01374-t002:** Comparison of genotype frequencies of *FSHR*, *INHA*, *ESR1*, and *BMP15* polymorphisms between the RIF and control subjects.

Genotypes	Controls (*n* = 317)	RIF (*n* = 133)	COR (95% CI)	*p*	FDR-*p*	AIC	AOR (95% CI)	*p*	FDR-*p*	AIC
*FSHR* rs6165 A>G										
AA	134 (42.3)	62 (46.6)	1.000 (reference)				1.000 (reference)			
AG	133 (42.0)	61 (45.9)	0.991 (0.647–1.520)	0.968	0.968	430.36	0.996 (0.650–1.528)	0.986	0.986	428.34
GG	50 (15.8)	10 (7.5)	0.432 (0.206–0.908)	0.027	0.135	282.93	0.463 (0.218–0.980)	0.044	0.165	280.03
Dominant (AA vs. AG+GG)			0.839 (0.558–1.260)	0.397	0.702	542.20	0.838 (0.558–1.260)	0.396	0.707	540.16
Recessive (AA+AG vs. GG)			0.434 (0.213–0.885)	0.022	0.11	540.73	0.430 (0.210–0.877)	0.020	0.100	538.69
HWE-*P*	0.083	0.340								
*INHA* rs11893842 A>G										
AA	106 (33.4)	42 (31.6)	1.000 (reference)				1.000 (reference)			
AG	156 (49.2)	61 (45.9)	0.987 (0.621–1.570)	0.956	0.968	518.96	0.988 (0.621–1.573)	0.961	0.986	518.96
GG	55 (17.4)	30 (22.6)	1.377 (0.778–2.436)	0.272	0.334	363.05	1.358 (0.767–2.407)	0.294	0.350	362.07
Dominant (AA vs. AG+GG)			1.089 (0.705–1.680)	0.702	0.702	540.03	1.086 (0.703–1.677)	0.710	0.710	539.99
Recessive (AA+AG vs. GG)			1.388 (0.842–2.287)	0.199	0.249	536.91	1.386 (0.841–2.286)	0.200	0.250	536.87
HWE-*P*	0.853	0.386								
*INHA* rs35118453 C>T										
CC	209 (65.9)	84 (63.2)	1.000 (reference)				1.000 (reference)			
CT	102 (32.2)	42 (31.6)	1.025 (0.660–1.590)	0.914	0.968	480.21	1.025 (0.660–1.592)	0.913	0.986	479.51
TT	6 (1.9)	7 (5.3)	2.903 (0.948–8.892)	0.062	0.155	292.70	2.865 (0.934–8.786)	0.066	0.165	286.68
Dominant (CC vs. CT+TT)			1.129 (0.740–1.722)	0.574	0.702	539.63	1.126 (0.738–1.718)	0.583	0.710	539.58
Recessive (CC+CT vs. TT)			2.880 (0.949–8.737)	0.062	0.155	534.27	2.875 (0.947–8.724)	0.062	0.155	534.09
HWE-*P*	0.106	0.564								
*ESR1* rs9340799 A>G										
AA	212 (66.9)	83 (62.4)	1.000 (reference)				1.000 (reference)			
AG	94 (29.7)	41 (30.8)	1.114 (0.713–1.740)	0.635	0.968	510.37	1.117 (0.715–1.745)	0.627	0.986	510.09
GG	11 (3.5)	9 (6.8)	2.090 (0.836–5.227)	0.115	0.192	247.08	2.107 (0.841–5.278)	0.112	0.187	240.74
Dominant (AA vs. AG+GG)			1.216 (0.798–1.854)	0.363	0.702	539.52	1.218 (0.799–1.857)	0.360	0.710	539.46
Recessive (AA+AG vs. GG)			2.019 (0.817–4.993)	0.128	0.213	538.12	2.023 (0.818–5.003)	0.127	0.212	538.05
HWE-*P*	0.884	0.217								
*ESR1* rs2234693 T>C										
TT	123 (38.8)	44 (33.1)	1.000 (reference)				1.000 (reference)			
TC	144 (45.4)	65 (48.9)	1.262 (0.803–1.983)	0.313	0.968	445.72	1.258 (0.800–1.978)	0.320	0.986	445.09
CC	50 (15.8)	24 (18.0)	1.342 (0.739–2.436)	0.334	0.334	279.86	1.330 (0.732–2.417)	0.350	0.350	279.60
Dominant (TT vs. TC+CC)			1.283 (0.838–1.964)	0.252	0.702	539.02	1.281 (0.836–1.962)	0.255	0.710	538.98
Recessive (TT+TC vs. CC)			1.176 (0.688–2.008)	0.553	0.553	540.00	1.173 (0.687–2.005)	0.559	0.559	539.96
HWE-*P*	0.470	0.999								
*BMP15* rs17003221 C>T										
CC	288 (90.9)	124 (93.2)	1.000 (reference)				1.000 (reference)			
CT	29 (9.1)	9 (6.8)	0.721 (0.331–1.568)	0.409	0.968	539.63	0.718 (0.330–1.563)	0.404	0.986	539.56
TT	0 (0.0)	0 (0.0)	N/A	N/A	N/A	N/A	N/A	N/A	N/A	N/A
Dominant (CC vs. CT+TT)			0.721 (0.331–1.568)	0.409	0.702	539.63	0.718 (0.330–1.563)	0.404	0.710	539.56
Recessive (CC+CT vs. TT)			N/A	N/A	N/A	N/A	N/A	N/A	N/A	N/A
HWE-*P*	0.393	0.686								
*BMP15* rs3810682 C>G										
CC	305 (96.2)	129 (97.0)	1.000 (reference)				1.000 (reference)			
CG	12 (3.8)	4 (3.0)	0.788 (0.250–2.490)	0.685	0.968	540.18	0.780 (0.246–2.470)	0.672	0.986	540.11
GG	0 (0.0)	0 (0.0)	N/A	N/A	N/A	N/A	N/A	N/A	N/A	N/A
Dominant (CC vs. CG+GG)			0.788 (0.250–2.490)	0.685	0.702	540.18	0.780 (0.246–2.470)	0.672	0.710	540.11
Recessive (CC+CG vs. GG)			N/A	N/A	N/A	N/A	N/A	N/A	N/A	N/A
HWE-*P*	0.731	0.860								

Note: AOR was adjusted by age. RIF, recurrent implantation failure; COR, crude odds ratio; AOR, adjusted odds ratio; 95% CI, 95% confidence interval; FDR; false discovery rate; AIC, Akaike information criterion; HWE, Hardy-Weinberg equilibrium, N/A, not applicable.

**Table 3 biomedicines-11-01374-t003:** Comparison of genotype frequencies of *FSHR*, *INHA*, *ESR1*, and *BMP15* polymorphisms between the RIF and control subjects.

Genotypes	Controls	RIF ≥ 3	AOR (95% CI)	*p*	FDR-*p*	AIC	RIF ≥ 4	AOR (95% CI)	*p*	FDR-*p*	AIC
	(*n* = 317)	(*n* = 119)					(*n* = 89)				
*FSHR* rs6165 A>G											
AA	106 (33.4)	36 (30.3)	1.000 (reference)				28 (31.5)	1.000 (reference)			
AG	156 (49.2)	54 (45.4)	1.037 (0.665–1.617)	0.871	0.967	393.87	42 (47.2)	0.955 (0.582–1.566)	0.854	0.969	335.77
GG	55 (17.4)	29 (24.4)	0.412 (0.183–0.931)	0.033	0.165	261.48	19 (21.3)	0.392 (0.156–0.982)	0.046	0.115	213.89
Dominant (AA vs. AG+GG)			0.852 (0.558–1.303)	0.461	0.706	504.07		0.787 (0.491–1.261)	0.319	0.809	420.25
Recessive (AA+AG vs. GG)			0.373 (0.171–0.815)	0.013	0.065	501.78		0.372 (0.154–0.901)	0.029	0.073	419.62
*INHA* rs11893842 A>G											
AA	209 (65.9)	75 (63.0)	1.000 (reference)				55 (61.8)	1.000 (reference)			
AG	102 (32.2)	38 (31.9)	1.010 (0.619–1.649)	0.967	0.967	485.21	28 (31.5)	1.011 (0.589–1.734)	0.969	0.969	399.29
GG	6 (1.9)	6 (5.0)	1.534 (0.851–2.767)	0.155	0.194	335.63	6 (6.7)	1.284 (0.657–2.511)	0.465	0.465	277.85
Dominant (AA vs. AG+GG)			1.147 (0.727–1.811)	0.555	0.706	504.15		1.082 (0.652–1.794)	0.761	0.809	419.89
Recessive (AA+AG vs. GG)			1.536 (0.922–2.557)	0.099	0.165	501.57		1.292 (0.720–2.319)	0.391	0.391	415.63
*INHA* rs35118453 C>T											
CC	134 (42.3)	55 (46.2)	1.000 (reference)				43 (48.3)	1.000 (reference)			
CT	133 (42.0)	56 (47.1)	1.029 (0.651–1.626)	0.903	0.967	449.91	40 (44.9)	1.033 (0.617–1.727)	0.903	0.969	376.03
TT	50 (15.8)	8 (6.7)	2.771 (0.863–8.895)	0.087	0.194	265.38	6 (6.7)	3.793 (1.171–12.285)	0.026	0.115	226.80
Dominant (CC vs. CT+TT)			1.123 (0.724–1.744)	0.605	0.706	503.88		1.182 (0.725–1.925)	0.503	0.809	419.35
Recessive (CC+CT vs. TT)			2.749 (0.868–8.704)	0.086	0.165	497.08		3.745 (1.176–11.921)	0.025	0.073	414.45
*ESR1* rs9340799 A>G											
AA	212 (66.9)	76 (63.9)	1.000 (reference)				56 (62.9)	1.000 (reference)			
AG	94 (29.7)	35 (29.4)	1.042 (0.652–1.666)	0.863	0.967	475.82	26 (29.2)	1.054 (0.623–1.783)	0.844	0.969	392.70
GG	11 (3.5)	8 (6.7)	2.060 (0.796–5.328)	0.136	0.194	351.20	7 (7.9)	2.459 (0.907–6.664)	0.077	0.128	291.35
Dominant (AA vs. AG+GG)			1.146 (0.737–1.782)	0.546	0.706	504.06		1.196 (0.732–1.952)	0.475	0.809	419.83
Recessive (AA+AG vs. GG)			2.025 (0.793–5.173)	0.140	0.175	502.35		2.400 (0.900–6.397)	0.080	0.133	417.51
*ESR1* rs2234693 T>C											
TT	123 (38.8)	38 (31.9)	1.000 (reference)				29 (32.6)				
TC	144 (45.4)	59 (49.6)	1.318 (0.820–2.118)	0.254	0.967	413.24	42 (47.2)	1.227 (0.720–2.089)	0.452	0.969	338.94
CC	50 (15.8)	22 (18.5)	1.393 (0.748–2.594)	0.297	0.297	257.72	18 (20.2)	1.498 (0.762–2.945)	0.241	0.301	219.97
Dominant (TT vs. TC+CC)			1.342 (0.858–2.100)	0.197	0.706	502.73		1.303 (0.792–2.144)	0.298	0.809	419.24
Recessive (TT+TC vs. CC)			1.197 (0.688–2.082)	0.524	0.524	504.02		1.340 (0.736–2.442)	0.339	0.391	419.45
*BMP15* rs17003221 C>T											
CC	288 (90.9)	110 (92.4)	1.000 (reference)				82 (92.1)	1.000 (reference)			
CT	29 (9.1)	9 (7.6)	0.804 (0.369–1.755)	0.584	0.967	504.11	7 (7.9)	0.834 (0.352–1.977)	0.681	0.969	420.16
TT	0 (0.0)	0 (0.0)	N/A	N/A	N/A	N/A	0 (0.0)	N/A	N/A	N/A	N/A
Dominant (CC vs. CT+TT)			0.804 (0.369–1.755)	0.584	0.706	504.11		0.834 (0.352–1.977)	0.681	0.809	420.16
Recessive (CC+CT vs. TT)			N/A	N/A	N/A	N/A		N/A	N/A	N/A	N/A
*BMP15* rs3810682 C>G											
CC	305 (96.2)	115 (96.6)	1.000 (reference)				86 (96.6)	1.000 (reference)			
CG	12 (3.8)	4 (3.4)	0.853 (0.269–2.709)	0.788	0.967	504.35	3 (3.4)	0.853 (0.234–3.104)	0.809	0.969	420.28
GG	0 (0.0)	0 (0.0)	N/A	N/A	N/A	N/A	0 (0.0)	N/A	N/A	N/A	N/A
Dominant (CC vs. CG+GG)			0.853 (0.269–2.709)	0.788	0.788	504.35		0.853 (0.234–3.104)	0.809	0.809	420.28
Recessive (CC+CG vs. GG)			N/A	N/A	N/A	N/A		N/A	N/A	N/A	N/A

Note: AOR was adjusted by age. RIF, recurrent implantation failure, AOR, adjusted odds ratio; 95% CI, 95% confidence interval; AIC, Akaike information criterion; N/A, not applicable.

**Table 4 biomedicines-11-01374-t004:** Combined genotype analysis for the *FSHR*, *INHA*, *ESR1*, and *BMP15* polymorphisms in RIF patients and controls.

Genotype Combinations	Controls	RIF	AOR (95% CI)	*p*
	(*n* = 317)	(*n* = 133)		
*FSHR* rs6165 A>G/*ESR1* rs9340799 A>G				
AA/AA	92 (29.0)	38 (28.6)	1.000 (reference)	
AA/AG	37 (11.7)	20 (15.0)	1.312 (0.673–2.556)	0.425
AA/GG	5 (1.6)	4 (3.0)	2.017 (0.487–8.352)	0.333
AG/AA	88 (27.8)	42 (31.6)	1.163 (0.685–1.974)	0.576
AG/AG	42 (13.2)	16 (12.0)	1.767 (0.278–11.232)	0.546
AG/GG	3 (0.9)	3 (2.3)	0.807 (0.270–2.410)	0.701
GG/AA	32 (10.1)	3 (2.3)	0.250 (0.072–0.874)	0.030
GG/AG	15 (4.7)	5 (3.8)	0.807 (0.270–2.410)	0.701
GG/GG	3 (0.9)	2 (1.5)	1.767 (0.278–11.232)	0.546
*INHA* rs35118453 C>T/*ESR1* rs2234693 T>C				
CC+CT/TT	119 (37.5)	42 (31.6)	1.000 (reference)	
CC+CT/TC+CC	192 (60.6)	84 (63.2)	1.240 (0.802–1.917)	0.333
TT/TT	4 (1.3)	2 (1.5)	1.403 (0.247–7.976)	0.703
TT/TC+CC	2 (0.6)	5 (3.8)	7.001 (1.298–37.776)	0.024
*FSHR* rs6165 A>G/*ESR1* rs9340799 A>G				
AA+AG/AA	180 (56.8)	80 (60.2)	1.000 (reference)	
AA+AG/AG+GG	87 (27.4)	43 (32.3)	1.115 (0.710–1.750)	0.636
GG/AA	32 (10.1)	3 (2.3)	0.199 (0.059–0.670)	0.009
GG/AG+GG	18 (5.7)	7 (5.3)	0.867 (0.348–2.160)	0.758
*FSHR* rs6165 A>G/*BMP15* rs17003221 C>T				
AA+AG/CC	241 (76.0)	114 (85.7)	1.000 (reference)	
AA+AG/CT+TT	26 (8.2)	9 (6.8)	0.723 (0.328–1.594)	0.421
GG/CC	47 (14.8)	10 (7.5)	0.447 (0.218–0.918)	0.028
GG/CT+TT	3 (0.9)	0 (0.0)	N/A	N/A
*FSHR* rs6165 A>G/*BMP15* rs3810682 C>G				
AA+AG/CC	257 (81.1)	119 (89.5)	1.000 (reference)	
AA+AG/CG+GG	10 (3.2)	4 (3.0)	0.842 (0.258–2.745)	0.775
GG/CC	48 (15.1)	10 (7.5)	0.446 (0.218–0.914)	0.027
GG/CG+GG	2 (0.6)	0 (0.0)	N/A	N/A
*INHA* rs11893842 A>G/*ESR1* rs9340799 A>G				
AA+AG/AA+AG	255 (80.4)	95 (71.4)	1.000 (reference)	
AA+AG/GG	7 (2.2)	8 (6.0)	3.065 (1.081–8.690)	0.035
GG/AA+AG	51 (16.1)	29 (21.8)	1.526 (0.913–2.549)	0.107
GG/GG	4 (1.3)	1 (0.8)	0.670 (0.074–6.073)	0.722
*FSHR* rs6165A>G/*INHA* rs35118453C>T				
AA+AG/CC+CT	261 (82.3)	117 (88.0)		
AA+AG/TT	6 (1.9)	6 (4.5)	2.203 (0.695–6.979)	0.180
GG/CC+CT	50 (15.8)	9 (6.8)	0.399 (0.190–0.841)	0.016
GG/TT	0 (0.0)	1 (0.8)	N/A	N/A
*FSHR* rs6165 A>G/*ESR1* rs9340799 A>G				
AA+AG/AA+AG	259 (81.7)	116 (87.2)	1.000 (reference)	
AA+AG/GG	8 (2.5)	7 (5.3)	1.979 (0.700–5.599)	0.198
GG/AA+AG	47 (14.8)	8 (6.0)	0.373 (0.171–0.816)	0.014
GG/GG	3 (0.9)	2 (1.5)	1.450 (0.239–8.817)	0.687
*FSHR* rs6165 A>G/*ESR1* rs2234693 T>C				
AA+AG/TT+TC	224 (70.7)	101 (75.9)	1.000 (reference)	
AA+AG/CC	43 (13.6)	22 (16.5)	0.604 (0.123–2.966)	0.686
GG/TT+TC	43 (13.6)	8 (6.0)	0.401 (0.181–0.886)	0.024
GG/CC	7 (2.2)	2 (1.5)	0.604 (0.123–2.966)	0.534
*FSHR* rs6165 A>G/*BMP15* rs17003221 C>T				
AA+AG/CC+CT	267 (84.2)	123 (92.5)	1.000 (reference)	
GG/CC+CT	50 (15.8)	10 (7.5)	0.430 (0.210–0.877)	0.020
*FSHR* rs6165 A>G/*BMP15* rs3810682 C>G				
AA+AG/CC+CG	267 (84.2)	123 (92.5)	1.000 (reference)	
GG/CC+CG	50 (15.8)	10 (7.5)	0.430 (0.210–0.877)	0.020
*ESR1* rs9340799 A>G/*ESR1* rs2234693 T>C				
AA+AG/TT+TC	266 (83.9)	104 (78.2)	1.000 (reference)	
AA+AG/CC	40 (12.6)	20 (15.0)	1.270 (0.709–2.276)	0.422
GG/TT+TC	1 (0.3)	5 (3.8)	12.930 (1.492–112.052)	0.020
GG/CC	10 (3.2)	4 (3.0)	1.024 (0.314–3.340)	0.968

Note: AOR was adjusted by age. RIF, recurrent implantation failure; 95% CI, 95% confidence interval; AOR, adjusted odds ratio; NA, not applicable.

**Table 5 biomedicines-11-01374-t005:** Clinical variables in RIF patients and controls stratified by *FSHR*, *INHA*, *ESR1*, and *BMP15* polymorphisms status by ANOVA and Kruskal–Wallis test.

Genotypes	BMI (kg/m^2^)	Homocysteine (μmol/L)	Folate (ng/mL)	BUN (mg/dL)	Creatinine (mg/dL)	Uric Acid (mg/dL)	Total Cholesterol (mg/dL)	TSH (mU/L)
	Mean ± SD	Mean ± SD	Mean ± SD	Mean ± SD	Mean ± SD	Mean ± SD	Mean ± SD	Mean ± SD
*FSHR* rs6165 A>G								
AA	21.4 ± 3.6	7.0 ± 2.3	16.8 ± 11.8	9.2 ± 3.0	0.7 ± 0.1	3.9 ± 1.0	199.1 ± 53.0	1.9 ± 1.4
AG	21.5 ± 3.2	6.3 ± 1.8	13.3 ± 6.2	9.5 ± 2.9	0.7 ± 0.2	3.9 ± 1.0	209.0 ± 55.6	1.9 ± 1.2
GG	21.5 ± 2.4	7.0 ± 3.8	9.0 ± 4.0	9.5 ± 2.9	0.7 ± 0.2	4.0 ± 1.2	202.0 ± 52.2	2.3 ± 1.2
*p* ^b^	0.676	0.273	0.175	0.533	0.576	0.987	0.366	0.209
*INHA* rs11893842 A>G								
AA	21.4 ± 3.2	6.4 ± 1.8	13.9 ± 8.7	9.5 ± 3.0	0.7 ± 0.2	4.0 ± 1.0	205.2 ± 57.2	2.1 ± 1.6
AG	21.7 ± 3.6	6.9 ± 2.5	16.2 ± 12.2	9.3 ± 2.9	0.7 ± 0.1	3.8 ± 1.0	205.6 ± 53.3	1.7 ± 1.0
GG	20.9 ± 2.6	7.0 ± 2.4	13.9 ± 5.5	9.4 ± 2.8	0.7 ± 0.2	4.0 ± 1.1	198.7 ± 52.1	1.9 ± 1.3
*p* ^b^	0.77	0.576 ^a^	0.838	0.873	0.789	0.381	0.532	0.332
*INHA* rs35118453 C>T								
CC	21.3 ± 3.1	6.6 ± 1.9	14.7 ± 10.7	9.4 ± 3.1	0.7 ± 0.2	3.9 ± 1.1	203.1 ± 56.5	2.0 ± 1.4
CT	21.8 ± 3.9	7.1 ± 2.8	14.4 ± 7.8	9.2 ± 2.7	0.7 ± 0.2	4.0 ± 0.9	206.6 ± 51.4	1.8 ± 1.0
TT	19.8 ± 1.7	6.3 ± 1.3	18.3 ± 6.5	9.9 ± 2.2	0.7 ± 0.2	3.6 ± 1.0	196.8 ± 34.7	2.1 ± 1.3
*p* ^b^	0.317	0.716	0.389	0.559	0.746	0.531	0.785	0.789
*ESR1* rs9340799 A>G								
AA	21.6 ± 3.6	6.7 ± 1.9	14.3 ± 7.6	9.4 ± 3.0	0.7 ± 0.2	3.9 ± 1.0	200.7 ± 51.7	1.8 ± 1.2
AG	21.0 ± 2.6	7.3 ± 3.0	17.9 ± 14.6	9.3 ± 2.8	0.7 ± 0.2	3.9 ± 1.1	214.5 ± 59.5	2.1 ± 1.4
GG	22.3 ± 4.2	5.6 ± 2.7	11.1 ± 7.7	9.7 ± 2.9	0.7 ± 0.1	4.1 ± 0.8	184.5 ± 42.4	2.2 ± 1.3
*p* ^b^	0.662	0.276 ^a^	0.502	0.847	0.726	0.618	0.116	0.42
*ESR1* rs2234693 T>C								
TT	21.8 ± 4.1	6.5 ± 1.6	12.9 ± 6.0	9.0 ± 3.0	0.7 ± 0.1	4.0 ± 1.2	204.8 ± 51.0	1.9 ± 1.4
TC	21.2 ± 2.7	6.8 ± 2.2	15.4 ± 11.4	9.5 ± 3.0	0.7 ± 0.2	3.8 ± 0.9	205.5 ± 58.8	1.9 ± 1.3
CC	21.5 ± 3.0	7.3 ± 3.4	18.3 ± 10.9	9.9 ± 2.5	0.7 ± 0.2	4.0 ± 1.0	197.6 ± 45.9	1.8 ± 1.0
*p* ^b^	0.974	0.456	0.294	0.085	0.76	0.92	0.872	0.927
*BMP15* rs17003221 C>T								
CC	21.4 ± 3.3	6.7 ± 2.3	15.1 ± 9.7	9.4 ± 3.0	0.7 ± 0.2	3.9 ± 1.1	204.5 ± 55.1	1.9 ± 1.3
CT	22.4 ± 4.0	7.5 ± 1.0	8.5 ± 1.0	9.1 ± 2.3	0.7 ± 0.2	3.9 ± 0.8	199.4 ± 44.7	1.8 ± 1.0
TT	0.0 ± 0.0	0.0 ± 0.0	0.0 ± 0.0	0.0 ± 0.0	0.0 ± 0.0	0.0 ± 0.0	0.0 ± 0.0	0.0 ± 0.0
*p* ^b^	0.243 ^a^	0.238	0.149	0.832	0.444	0.902	0.977	0.94
*BMP15* rs3810682 C>G								
CC	21.5 ± 3.3	6.7 ± 2.3	15.0 ± 9.7	9.4 ± 3.0	0.7 ± 0.2	3.9 ± 1.0	203.8 ± 54.3	1.9 ± 1.3
CG	20.6 ± 3.5	7.1 ± 1.3	10.9 ± 3.9	9.2 ± 1.8	0.7 ± 0.2	4.0 ± 0.9	208.4 ± 51.4	1.6 ± 1.1
GG	0.0 ± 0.0	0.0 ± 0.0	0.0 ± 0.0	0.0 ± 0.0	0.0 ± 0.0	0.0 ± 0.0	0.0 ± 0.0	0.0 ± 0.0
*p* ^b^	0.303 ^a^	0.636	0.537	0.909	0.928	0.674	0.785	0.741
**Genotypes**	**E2 (pg/mL)**	**FSH (U/L)**	**LH (U/L)**	**WBC (10^3^/μL)**	**Hgb (g/dL)**	**PLT (10^3^/μL)**	**PT (sec)**	**aPTT (sec)**
	**Mean ± SD**	**Mean ± SD**	**Mean ± SD**	**Mean ± SD**	**Mean ± SD**	**Mean ± SD**	**Mean ± SD**	**Mean ± SD**
*FSHR* rs6165 A>G								
AA	29.0 ± 14.4	8.5 ± 4.4	4.2 ± 2.4	7.9 ± 2.6	12.4 ± 1.3	235.6 ± 62.3	11.1 ± 1.8	29.5 ± 3.5
AG	33.0 ± 16.8	8.2 ± 3.2	4.3 ± 2.9	7.9 ± 2.5	12.4 ± 1.4	231.1 ± 60.5	10.8 ± 0.9	29.3 ± 3.5
GG	25.9 ± 12.9	8.8 ± 2.2	4.0 ± 1.5	6.8 ± 2.1	12.0 ± 1.5	234.0 ± 89.9	10.7 ± 1.0	28.4 ± 2.8
*p* ^b^	0.076	0.11	0.843	0.119	0.429	0.576	0.207	0.334
*INHA* rs11893842 A>G								
AA	27.2 ± 13.3	8.6 ± 3.5	3.9 ± 2.0	7.6 ± 2.3	12.4 ± 1.2	236.4 ± 73.8	10.8 ± 0.8	29.3 ± 3.9
AG	30.4 ± 16.1	8.1 ± 4.2	4.1 ± 2.9	7.9 ± 2.7	12.3 ± 1.4	233.5 ± 64.7	11.0 ± 1.8	29.1 ± 3.2
GG	34.0 ± 15.7	9.0 ± 1.7	4.7 ± 2.2	7.6 ± 2.2	12.3 ± 1.4	228.2 ± 47.5	10.8 ± 0.8	29.7 ± 3.4
*p* ^b^	0.067	0.113	0.077	0.682	0.812	0.97	0.788	0.521
*INHA* rs35118453 C > T								
CC	29.0 ± 15.0	8.5 ± 4.0	4.1 ± 2.7	7.6 ± 2.4	12.3 ± 1.3	236.6 ± 67.6	10.9 ± 0.9	29.3 ± 3.5
CT	31.0 ± 15.4	8.4 ± 3.0	4.3 ± 1.9	7.9 ± 2.6	12.4 ± 1.4	229.0 ± 60.4	11.0 ± 2.0	29.3 ± 3.5
TT	44.9 ± 15.0	7.1 ± 2.8	6.1 ± 1.7	8.6 ± 2.5	12.6 ± 1.6	217.6 ± 50.1	10.7 ± 0.6	29.8 ± 2.8
*p* ^b^	0.067	0.066	0.017	0.232	0.262	0.482	0.878	0.634
*ESR1* rs9340799 A>G								
AA	30.5 ± 16.2	8.5 ± 4.0	4.3 ± 2.7	7.8 ± 2.5	12.4 ± 1.4	234.6 ± 69.6	10.9 ± 0.8	29.2 ± 3.4
AG	28.8 ± 13.4	8.4 ± 3.0	4.0 ± 2.1	7.8 ± 2.5	12.4 ± 1.3	233.3 ± 55.8	11.0 ± 2.2	29.4 ± 3.6
GG	30.2 ± 12.6	7.1 ± 2.8	3.3 ± 1.1	6.9 ± 1.9	12.0 ± 1.2	219.8 ± 48.4	11.1 ± 1.0	29.7 ± 3.7
*p* ^b^	0.882	0.267	0.576	0.423	0.381	0.644	0.51	0.918
*ESR1* rs2234693 T>C								
TT	28.7 ± 14.2	8.4 ± 3.6	4.1 ± 3.0	7.9 ± 2.6	12.4 ± 1.5	232.3 ± 59.5	10.9 ± 0.9	29.3 ± 3.7
TC	30.9 ± 15.5	8.1 ± 3.3	4.3 ± 2.3	7.7 ± 2.5	12.4 ± 1.2	237.7 ± 71.0	11.0 ± 1.8	29.4 ± 3.4
CC	30.0 ± 16.8	9.2 ± 4.7	4.0 ± 1.8	7.7 ± 2.3	12.2 ± 1.3	223.1 ± 55.8	10.8 ± 0.8	29.0 ± 3.1
*p* ^b^	0.58	0.446	0.601	0.861	0.761	0.543	0.775	0.625
*BMP15* rs17003221 C > T								
CC	30.1 ± 15.0	8.5 ± 3.7	4.2 ± 2.5	7.8 ± 2.6	12.4 ± 1.4	233.2 ± 64.2	10.9 ± 1.4	29.4 ± 3.5
CT	27.0 ± 19.8	7.8 ± 2.5	4.0 ± 0.9	7.7 ± 1.6	12.0 ± 1.0	234.9 ± 69.6	10.8 ± 0.8	28.5 ± 3.1
TT	0.0 ± 0.0	0.0 ± 0.0	0.0 ± 0.0	0.0 ± 0.0	0.0 ± 0.0	0.0 ± 0.0	0.0 ± 0.0	0.0 ± 0.0
*p* ^b^	0.55	0.5	0.524	0.68	0.039	0.812	0.553	0.146
*BMP15* rs3810682 C>G								
CC	29.7 ± 15.2	8.4 ± 3.7	4.2 ± 2.5	7.7 ± 2.5	12.3 ± 1.3	233.4 ± 65.1	10.9 ± 1.4	29.3 ± 3.4
CG	35.6 ± 15.6	8.4 ± 3.4	4.4 ± 1.5	8.7 ± 2.2	12.4 ± 1.8	231.9 ± 53.6	10.6 ± 0.6	30.0 ± 5.2
GG	0.0 ± 0.0	0.0 ± 0.0	0.0 ± 0.0	0.0 ± 0.0	0.0 ± 0.0	0.0 ± 0.0	0.0 ± 0.0	0.0 ± 0.0
*P^b^*	0.248	0.902	0.34	0.105	0.68	0.732	0.363	0.761

ANOVA, Analysis of variance; BMI, body mass index; BUN, blood urea nitrogen; TSH, thyroid stimulating hormone; E2, estradiol; FSH, follicle stimulating hormone; LH, luteinizing hormone; WBC, white blood cell; Hgb, hemoglobin; PLT, platelet; PT, prothrombin time; aPTT, activated partial thromboplastin time; SD, standard deviation. ^a^
*p*-values were calculated by ANOVA. ^b^
*p*-values were calculated by Kruskal–Wallis tests.

## Data Availability

The data presented in this study are available on request from the corresponding author.

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
