# Peer review of "Association of Polymorphisms in FSHR, INHA, ESR1, and BMP15 with Recurrent Implantation Failure"

_biomedicines, 2023, doi:10.3390/biomedicines11051374_

Round 1

Reviewer 1 Report

The manuscript submitted to the Biomedicines by Dr. Eun Ju Ko et al. and entitled «Association of polymorphisms in FSHR, INHA, ESR1, and BMP15 with recurrent implantation failure» is described the case-control study aimed to investigation of the role of genetic polymorphism in the recurrent implantation failure. The study is well designed, performed in the sufficient samole size and the aprropriate statistical analysis was performed. The are some issues that must be solved before manuscript can be published.

1. Criteria of SNP selection must be added to the Matherial and Methods section.

2. Please add the Akaike’s information criterion (AIC) in the Table 2 and 3 to discover the most likely inheritance model for each specific gene polymorphism.

3. Why authors didn't apply the FDR coorrection while genotype frequencies was studied among patients according to the number of RIFs (Table 3)? Authors must applied this correction in this analysis.

4. How combined genotype analysis was performed? What tools authors used for this analysis (MDR or another)? Please incude this information in the Matherial and methods section.

Author Response

The manuscript submitted to the Biomedicines by Dr. Eun Ju Ko et al. and entitled «Association of polymorphisms in FSHR, INHA, ESR1, and BMP15 with recurrent implantation failure» is described the case-control study aimed to investigation of the role of genetic polymorphism in the recurrent implantation failure. The study is well designed, performed in the sufficient sample size and the appropriate statistical analysis was performed. There are some issues that must be solved before manuscript can be published.
=> Thank you for critical comments. We tried to augment some contents as you suggested. We revised our insufficient descriptions as follows:

  1. Criteria of SNP selection must be added to the Material and Methods section.

=> Thank you for comment. We agree for your comments. So, we added the following paragraph to the material and method section.

“We selected FSHR, INHA, ESR1, and BMP15, which are hormone-related genes associated with pregnancy. To select polymorphisms of the FSHR, INHA, ESR1, and BMP15 genes, studies on the association between pregnancy-related diseases (recurrent pregnancy loss, recurrent implantation failure, preeclampsia, premature ovarian failure and poor ovarian response) and polymorphisms were investigated [46, 50-55]. Finally, a total of 7 polymorphisms in FSHR (rs6165), INHA (rs11893842 and rs35118453), ESR1 (rs9340799 and rs2234693), and BMP15 (rs17003221 and rs3810682) were selected and studied.” [Material and Methods section, page5, line209-216]

  1. Please add the Akaike’s information criterion (AIC) in the Table 2 and 3 to discover the most likely inheritance model for each specific gene polymorphism.

=> Thank you for comment. We agree for your comments. So, we added the AIC in the table 2,3 and added the following paragraph to the material and method section.

“To select the best inheritance model for a specific polymorphism, Akaike's information criterion was calculated.” [Material and Methods section, page 5, line 235-236]

  1. Why authors didn't apply the FDR correction while genotype frequencies was studied among patients according to the number of RIFs (Table 3)? Authors must applied this correction in this analysis.

=> Thank you for comment. We agree for your comments. So, we added the FDR-P in the table 3.

  1. How combined genotype analysis was performed? What tools authors used for this analysis (MDR or another)? Please include this information in the Material and methods section.

=> Thank you for comment. So, we added the following paragraph to the material and method section.

“The open source MDR software package (v.2.0, www.www.epistasis.org) was used to perform genetic interaction analysis. Using this MDR analysis, all possible genotype combinations for gene-gene interactions were identified and analyzed.” [Material and Methods section, pag64, line241-243]

Reviewer 2 Report

Thank you a well presented and clear manuscript. I have a couple of minor suggestions. The reasoning for the choice of genes in which to examine polymorphisms was not made clear in the introduction but mentioned on lines 322 to 325 of the discussion. It might clarify things to move or repeat the statement. On line 69 there begins a mention of the role of FSHR in relation to murine pregnancy. Is it known how this might be important in the human condition in the same manner? 

It would be interesting to know how the work is to be continued and whether it might affect testing in Korea.

Author Response

Thank you a well presented and clear manuscript. I have a couple of minor suggestions. The reasoning for the choice of genes in which to examine polymorphisms was not made clear in the introduction but mentioned on lines 322 to 325 of the discussion. It might clarify things to move or repeat the statement. On line 69 there begins a mention of the role of FSHR in relation to murine pregnancy. Is it known how this might be important in the human condition in the same manner? It would be interesting to know how the work is to be continued and whether it might affect testing in Korea.

=> Thank you for critical comments. We tried to augment some contents as you suggested. We revised our insufficient descriptions as follows:

  1. The reasoning for the choice of genes in which to examine polymorphisms was not made clear in the introduction but mentioned on lines 322 to 325 of the discussion. It might clarify things to move or repeat the statement.

=> Thank you for comment. We agree for your comments. So, we added the following paragraph to the introduction section.

“Polymorphisms in regulatory regions (promoter, 5', 3' UTR) or gene-coding regions may alter gene expression [40, 41]. Maternal hormones play an important role in maintaining pregnancy, and hormone levels may be altered by certain polymorphisms [42, 43]. Therefore, four genes (FSHR, INHA, ESR1, BMP15) related to hormones were selected, and polymorphisms located in gene regulatory or coding regions were selected Finally, A total of 7 mutations were selected: coding region (FSHR rs6165, ESR1 rs2234693, BMP15 rs17003221), promoter region (INHA rs11893842, rs35118453 and ESR1 rs9340799), and 5'UTR region (BMP15 rs3810682).” [Introduction section, page3, line 106-114]                      

  1. On line 69 there begins a mention of the role of FSHR in relation to murine pregnancy. Is it known how this might be important in the human condition in the same manner?

=> Thank you for comment. We agree for your comments. So, we added the added the following paragraph to the introduction section.

“Extraovarian FSHRs play an important role in establishing and maintaining successful pregnancies in humans [22]. Previously reported studies showed that FSHR was expressed in fetal vascular endothelium [23], and it was identified that the FSH-FSHR signaling system promotes the angiogenesis of vascular endothelial cells [25]. Also, FSHR is also expressed in uterine myometrium and plays an important role in regulating uterine muscle contraction [26].” [Introduction section, page2, line 77-82]